# Molecular Genetic Diversity and Line × Tester Analysis for Resistance to Late Wilt Disease and Grain Yield in Maize

Mohamed M. Kamara [1], Nasr A. Ghazy [2], Elsayed Mansour [3,*], Mohsen M. Elsharkawy [4], Ahmed M. S. Kheir [5] and Khaled M. Ibrahim [6]

1   Department of Agronomy, Faculty of Agriculture, Kafrelsheikh University, Kafr El-Sheikh 33516, Egypt; mohamed.kamara@agr.kfs.edu.eg
2   Maize and Sugar Crops Diseases Department, Plant Pathology Research Institute, Agricultural Research Center, Giza 12112, Egypt; nasrghazy@yahoo.com
3   Department of Crop Science, Faculty of Agriculture, Zagazig University, Zagazig 44519, Egypt
4   Department of Agricultural Botany, Faculty of Agriculture, Kafrelsheikh University, Kafr Elsheikh 33516, Egypt; mohsen.abdelrahman@agr.kfs.edu.eg
5   Soils, Water and Environment Research Institute, Agricultural Research Center, Giza 12619, Egypt; ahmed.kheir@arc.sci.eg
6   Agronomy Department, Faculty of Agriculture, New Valley University, El-Kharga 72511, Egypt; K.elgeady@agr.nvu.edu.eg
*   Correspondence: sayed_mansour_84@yahoo.es

**Abstract:** Late wilt disease (LWD) caused by the fungus *Magnaporthiopsis maydis* poses a major threat to maize production. Developing high-yielding and resistant hybrids is vital to cope with this destructive disease. The present study aimed at assessing general (GCA) and specific (SCA) combining abilities for agronomic traits and resistance to LWD, identifying high-yielding hybrids with high resistance to LWD, determining the parental genetic distance (GD) using SSR markers and investigating its relationship with hybrid performance and SCA effects. Ten diverse yellow maize inbred lines assembled from different origins and three high-yielding testers were crossed using line × tester mating design. The obtained 30 test-crosses plus the check hybrid TWC-368 were evaluated in two field trials. Earliness and agronomic traits were evaluated in two different locations. While resistance to LWD was tested under two nitrogen levels (low and high levels) in a disease nursery that was artificially infected by the pathogen *Magnaporthiopsis maydis*. Highly significant differences were detected among the evaluated lines, testers, and their corresponding hybrids for most measured traits. The non-additive gene action had more important role than the additive one in controlling the inheritance of earliness, grain yield, and resistance to LWD. The inbred lines L4 and L5 were identified as an excellent source of favorable alleles for high yielding and resistance to LWD. Four hybrids L5 × T1, L9 × T1, L4 × T2, and L5 × T2, exhibited earliness, high grain yield, and high resistance to LWD. Parental GD ranged from 0.60 to 0.97, with an average of 0.81. The dendrogram grouped the parental genotypes into three main clusters, which could help in reducing number of generated crosses that will be evaluated in field trials. SCA displayed significant association with the hybrid performance for grain yield and resistance to LWD, which suggests SCA is a good predictor for grain yield and resistance to LWD.

**Keywords:** *Magnaporthiopsis maydis*; *Zea mays*; high-yielding hybrids; SSR markers; combining ability

## 1. Introduction

Maize (*Zea mays* L.) is the third most prominent cereal crop after wheat and rice [1,2]. Global climate change and fluctuation in environmental conditions are expected to cause several biotic stresses which adversely affect maize growth and productivity [3–6]. Among biotic stresses, LWD caused by the fungus *Magnaporthiopsis maydis* [7] is one of the major devastating diseases [8]. This disease is generally characterized by rapid wilting of maize plants prior to the physiological maturity stage. It colonizes xylem and initially appears

in the leaves during the tasseling stage and then develops into the stalks [7]. Significant yield loss due to LWD was reported in different countries including Spain and Portugal [9], India [10], and Egypt [11]. It poses a serious hazard to maize production, with a degree of yield loss that may reach up to 50% in the infested fields [12]. Chemical control of the disease is partially effective [13] or still under development [14]. Thus, developing hybrids for resistance to LWD seems to be the best approach to reduce yield loss for smallholder farmers [15–17]. However, the majority of resistant hybrids to LWD are low-yielding or have other undesirable agronomic characteristics. Therefore, breeding for resistance to LWD is a crucial goal to attenuate yield losses and sustain food security. Thereupon, identifying hybrid combinations that contribute to both high grain yield and resistance to LWD is essential in maize breeding. Moreover, understanding the genetic diversity and mode of gene action for grain yield and resistance to LWD is important for breeding high-yielding and resistant maize hybrids [18,19].

Developing high-yielding hybrids and resistant to LWD depends rigidly on the correct choice of parents [20,21]. Line × tester mating scheme is an effective method to estimate general (GCA) and specific (SCA) combining ability effects and recognizes the appropriate parents. Moreover, it identifies gene action that is responsible for the expression of the interest traits even in small sample size [22–24]. This method helps in selecting superior parents for developing high-yielding hybrids. The GCA and SCA variances are used to determine the contribution of additive and non-additive gene effects concerned in the expression of targeted characteristics [25,26]. The GCA represents additive gene effects, while SCA refers to the deviation of hybrid performance from the parents used, and it is associated with non-additive gene effects [23,27]. The additive genetic effect was depicted to be valuable in resistance to LWD [28], whereas non-additive gene action has a greater role in the genetic control of maize grain yield under different environments [2,25,26]. However, little information is available regarding resistance to LWD alongside high yielding in maize.

Knowledge of genetic distance (GD) is of great importance to maize breeders [29]. It accelerates discovering successful hybrids without evaluating all possible parent combinations in maize breeding programs [30]. Recently, a vast number of DNA markers proved successful for unfolding GD of maize inbred lines. Simple sequence repeats (SSRs) are among the most important molecular markers. SSRs are co-dominant, multi-allelic, highly informative, and reproducible markers [31,32]. The parental GD using molecular markers is employed to predict hybrid performance and SCA effects in maize [27,29,30]. Several studies detected a significant association between GD-based molecular markers and F1 hybrid performance in maize [33–35], although, other studies reported no association between GD and F1 hybrids [2,27,34]. Consequently, the potential of molecular markers in determining the extent of hybrid performance and SCA in maize is inconclusive. Thus, the present study was undertaken to (1) determine GCA of the assembled inbred lines and testers as well as SCA of their corresponding hybrids; (2) elucidate the type of gene action controlling yield traits and resistance to LWD; (3) identify high-yielding and LWD resistant hybrids; and (4) assess the parental genetic distance and its relationship with hybrid performance and SCA.

## 2. Materials and Methods

### 2.1. Plant Materials

Ten yellow maize (*Zea mays* L.) inbred lines were used in this study. Four inbreds; IL185 (L1), IL176 (L2), IL202 (L3), and IL203 (L4) were obtained from Maize Research Department, Agricultural Research Center (ARC), Egypt, while the other inbreds; CML217 (L5), CML224 (L6), CML225 (L7), CML226 (L8), CML228 (L9), and CML289 (L10) were introduced from the International Maize and Wheat Improvement Center (CIMMYT). The name, pedigree, and source of these inbred lines are shown in Supplementary Materials Table S1. The 10 inbred lines were crossed with three high-yielding commercial hybrids; SC-168 (T1), SC-176 (T2) and TWC-352 (T3) using line × tester mating design during the summer of 2018 at the Experimental Farm, Faculty of Agriculture, Kafrelsheikh University, Egypt.

### 2.2. Yield Trials

The obtained 30 test-crosses and newly developed high-yielding commercial check hybrid TWC-368 were evaluated at two different locations (Sakha and El-Kharga) during the growing season of 2019. The first location was Sakha Agricultural Research Station, Agricultural Research Center (ARC), Egypt (30°3′ N, 31°3′ E) and the second one was the Experimental Farm, Faculty of Agriculture, New Valley University, El-Kharga, Egypt (30°19′ N, 25°15′ E). The two locations; Sakha and El-Kharga represented distinct soil types. The soil in Sakha represents the old Nile valley soils and is classified as clay soil (24.32% sand, 23.99% silt, and 51.69%clay) while the soil in El-Kharga represents the newly reclaimed soils and classified as sandy soil (86.50% sand, 8.58% silt, and 4.92% clay). The physical and chemical soil properties are presented in Supplementary Materials Table S2. Additionally, the meteorological data in each location are presented in Figure 1. Randomized Complete Block Design (RCBD) with three replications was applied in each location. Each plot consisted of 2-ridges of 5-m long and 0.7-m width. With two kernels/hill on one side of the ridge, the hills were spaced at 0.25-m, later thinned to one plant/hill. Standard agronomic practices including insect pest and weed control were applied as recommended for growing maize in the region.

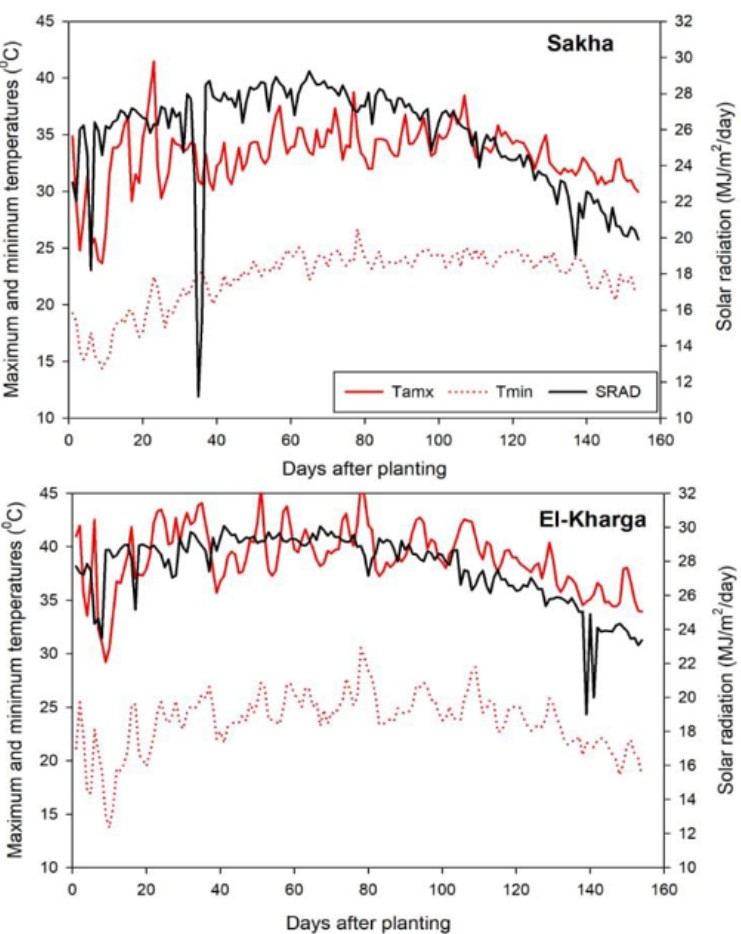

**Figure 1.** Daily maximum (T max) and minimum temperature (T min) as well as solar radiation (SRAD) for the two used locations.

### 2.3. Late Wilt Trial

The obtained 30 test-crosses and the check hybrid TWC-368 were evaluated in the disease nursery under artificial soil inoculation by the pathogen, *Magnaporthiopsis maydis*, under two levels of nitrogen fertilizer, i.e., 144 kg ha$^{-1}$ (low) and 288 kg ha$^{-1}$ (high). Annually, the used nursery is infected artificially by the pathogen *Magnaporthiopsis maydis*

(4 clonal lineages) according to Zeller et al. [36] to improve selection efficiency and distinguishing resistant and susceptible genotypes. The trial was performed for each nitrogen level (separate experiments) using RCBD with three replications at Sakha Agricultural Research Station during the summer growing season of 2019. The plot size was one row, 5-m long, 0.7-m wide, and 0.25-m between hills. The nitrogen fertilizer was added in two equal doses before the first and second irrigation. The rest of recommended agricultural practices was applied.

### 2.4. Data Collection

In the yield trial, data were recorded on days to 50% silking (DTS), ear length (EL), ear diameter (ED), number of rows/ear (NRPE), number of kernels/row (NKPR), hundred kernel weight (HKW), and grain yield/plant (GYPP) for the evaluated hybrids in the yield trials. DTS was recorded when half of the plants in each plot began forming silks. At harvest, 10 random ears were picked from each plot to measure EL, ED, NRPE, NKPR, and HKW. Plots were hand-harvested and GYPP was measured by using weight of the shelled grain (adjusted to 15.5% grain moisture content). A hand-held moisture meter was utilized to estimate grain moisture at harvest time. Data were recorded in the LWD trial after 35 days of 50% silking according to El-Shafey et al. [37]. The infected plants from each plot were recorded and used to calculate the percentage of resistance to LWD as follows:

$$\text{Resistance to LWD \%} = \left( \frac{\text{No. of uninfected plants in each plot}}{\text{No. of total plants in each plot}} \right) \times 100$$

### 2.5. Statistical Analysis

Analysis of variance (ANOVA) was applied for all obtained data using R software version 3.6.1. Combined analysis was performed across the two locations and the two nitrogen levels where the homogeneity test was non-significant. The angular transformation was used for resistance to LWD percentage as outlined by Snedecor and Cochran [38]. To identify the significance of variations between means, the least significant difference (LSD) values were estimated. The GCA effects of the lines and testers and SCA effects of the hybrids were calculated using line × tester analysis according to Kempthorne [22].

### 2.6. Molecular Analysis

Ten seeds of each genotype (10 lines and 3 testers) were grown into labeled pots. Genomic DNA was extracted using the CTAB method [39] from healthy portion of young leaves collected from 20-day-old seedlings. Quantity and consistency of DNA were measured using the NanoDrop spectrophotometer (ND-1000, USA). Twenty-four SSR primer pairs were analyzed. The sequence information of each primer pair was obtained from the MaizeGDB database (www.maizegdb.org, accessed on 15 February 2021) and it showed in Supplementary Materials Table S3. PCR was conducted using a 10 μL reaction volume comprising 1 μL of 20 ng/μL genomic DNA template, 2 mM $MgCl_2$, 0.2 mM dNTPs, 1 Taq DNA polymerase unit (Promega, Madison, WI, USA) and 0.5 μM reverse and forward primers. The PCR reaction was operated by pre-denaturation at 94 °C for 2 min followed by 94 °C for 30 s, annealing at 55 °C for 30 s, 30 s of extension at 72 °C for 35 cycles and ending with 3 min of elongation at 72 °C. Amplification products were resolved in 2% agarose gel. The gel was stained with ethidium bromide then visualized under UV light and documented using a gel documentation system (UVITEC, Cambridge, UK). The amplified bands were graded for each SSR marker on the basis of presence or absence of the bands, producing a binary data matrix of (1) and (0). Allele number, major allele frequency, gene diversity, and polymorphic information content (PIC) were determined for each marker using PowerMarker version 3.25. Genetic distances of the 13 parental genotypes were estimated according to Jaccard [40] using the PAST program. Neighbor-joining trees were designed and displayed using the Darwin 6 software.

## 3. Results

### 3.1. Yield Trials

#### 3.1.1. ANOVA and Line × Tester Analysis

The combined ANOVA showed a significant difference ($p < 0.05$) for locations (Loc), genotypes (G), crosses (C), and their interactions for most evaluated traits (Table 1). Moreover, significant variance ($p < 0.05$) due to lines (L), testers (T), and lines × testers (L × T) were detected for all measured traits, except ED and HKW for testers ($p > 0.05$). Additionally, the variance due to the interactions of L × Loc, T × Loc and L × T × Loc were also significant ($p < 0.01$) for all the measured traits except NRPE and GYPP for T × Loc (Table 1). The SCA variance was higher than GCA variance for all studied traits. Furthermore, the magnitude of SCA × Loc interaction was higher than those of GCA × Loc interaction for all evaluated traits (Table 1).

**Table 1.** Ordinary analysis of variance and line × tester analysis for grain yield and related agronomic traits across two locations.

| Source of Variance | DF | DTS | EL | ED | NRPE | NKPR | HKW | GYPP |
|---|---|---|---|---|---|---|---|---|
| | | | | Analysis of variance (mean squares) | | | | |
| Locations (Loc) | 1 | 503.42 ** | 20.47 * | 6.16 * | 19.96 * | 4594.37 ** | 4499.54 ** | 224161 ** |
| Replication (Location) | 4 | 3.31 | 2.47 | 0.30 | 1.71 | 15.71 | 11.55 | 58.66 |
| Genotypes (G) | 30 | 25.52 ** | 5.54 ** | 0.49 ** | 7.52 ** | 40.48 ** | 58.86 ** | 1952 ** |
| Crosses (C) | 29 | 25.59 ** | 5.66 ** | 0.50 ** | 7.54 ** | 41.85 ** | 60.72 ** | 2001 ** |
| C vs. Check | 1 | 23.61 ** | 1.94 | 0.19 | 6.94 ** | 0.79 | 5.01 | 525.8 ** |
| G × Loc | 30 | 15.10 ** | 11.48 ** | 0.61 ** | 3.83 ** | 41.59 ** | 54.22 ** | 2269 ** |
| C × Loc | 29 | 15.34 ** | 11.49 ** | 0.61 ** | 3.74 ** | 40.38 ** | 52.00 ** | 2329 ** |
| C vs. Chec × Loc | 1 | 8.13 | 11.33 ** | 0.68 ** | 6.55 * | 76.82 ** | 118.58 ** | 534.5 ** |
| Error | 120 | 2.11 | 0.86 | 0.05 | 0.97 | 4.26 | 7.57 | 49.56 |
| | | | | Line × tester analysis (mean squares) | | | | |
| Lines (L) | 9 | 61.19 ** | 8.83 ** | 0.65 ** | 15.55 ** | 52.68 ** | 132.09 ** | 3054 ** |
| Testers (T) | 2 | 9.65 * | 17.23 ** | 0.16 | 4.65 ** | 71.38 ** | 2.65 | 2884 ** |
| L × T | 18 | 9.56 ** | 2.79 ** | 0.46 ** | 3.85 ** | 33.16 ** | 31.49 ** | 1377 ** |
| L × Loc | 9 | 23.73 ** | 21.66 ** | 0.38 ** | 3.49 ** | 60.58 ** | 84.75 ** | 5042 ** |
| T × Loc | 2 | 44.45 ** | 18.37 ** | 1.26 ** | 1.99 | 118.73 ** | 127.17 ** | 32.55 |
| L × T × Loc | 18 | 7.91 ** | 5.64 ** | 0.65 ** | 4.06 ** | 21.57 ** | 27.27 ** | 1228 ** |
| Error | 116 | 2.14 | 0.87 | 0.05 | 0.99 | 4.38 | 7.64 | 51.08 |
| $K^2$ GCA | | 0.85 | 0.31 | 0.01 | 0.23 | 1.48 | 1.53 | 74.82 |
| $K^2$ SCA | | 1.24 | 0.32 | 0.07 | 0.48 | 4.80 | 3.98 | 220.99 |
| $K^2$ GCA × Loc | | 1.64 | 0.98 | 0.04 | 0.09 | 4.37 | 5.04 | 127.5 |
| $K^2$ SCA × Loc | | 1.92 | 1.59 | 0.20 | 1.02 | 5.73 | 6.54 | 392.2 |

* and ** indicate *p*-value < 0.05 and 0.01, respectively, DF: degree of freedom, DTS: days to 50% silking, EL: ear length, ED: ear diameter, NRPE: number of rows/ear, NKPR: number of kernels/row, HKW: hundred kernel weight and GYPP: grain yield/plant.

#### 3.1.2. Mean Performance

The performance of 30 test-crosses and check hybrid (TWC-368) for agronomic traits are illustrated in Table 2. The values of all traits differed significantly among the evaluated hybrids. The values of days to 50% silking (DTS) ranged from 59.33 (L10 × T2) to 67.17 days (L4 × T1) with an average of 63.48 days. A total of 18 hybrids were significantly earlier than the commercial check hybrid. The ear length (EL) average was 16.94 cm, and the longest value was observed in L3 × T2 (18.67 cm), while L9 × T1 recorded the shortest EL (14.52 cm). Furthermore, two crosses L3 × T2 and L2 × T3 significantly exceeded the check hybrid. Similarly, ear diameter (ED) varied from 3.85 cm (L3 × T1) to 4.95 (L1 × T2) with an average of 4.42 cm. Three crosses L1 × T2, L7 × T2 and L9 × T3 displayed significantly higher values than the check hybrid. The average of number of rows/ear (NRPE) was 14.87 ranging from 13.15 (L6 × T1) to 17.17 (L5 × T2). In the same context, the highest

number of kernels/row (NKPR) was assigned for L2 × T2 (38.8), whereas the lowest value was exhibited by L6 × T2 (29.15). The crosses L6 × T1, L8 × T1, L1 × T2, L2 × T2, L3 × T2, L4 × T2, and L5 × T2 possessed higher NKPR than the check hybrid. The average of hundred kernel weight (HKW) was 30.58 g, and the heaviest kernel index was obtained by the hybrid L5 × T3 (37.21 g), whereas L1 × T3 produced the lightest kernels (24.83 g). In addition, the three hybrids L5 × T1, L4 × T2 and L5 × T3 displayed higher HKW than the check hybrid. The average of grain yield/plant (GYPP) ranged from 100.00 g (L10 × T3) to 174.18 g (L5 × T2) with an average of 136.48 g. Obviously, the crosses L5 × T1, L9 × T1, L4 × T2, and L5 × T2 significantly surpassed the check hybrid by 10.41%, 6.98%, 11.07%, and 19.30%, respectively.

**Table 2.** Mean performance of the thirty hybrids and the check hybrid TW-368 for agronomic traits averaged over two locations.

| Hybrid | DTS | EL (cm) | ED (cm) | NRPE | NKPR | HKW (g) | GYPP (g) |
|---|---|---|---|---|---|---|---|
| L1 × T1 | 64.17 | 16.37 | 4.10 | 15.30 | 34.19 | 27.16 | 126.61 |
| L2 × T1 | 62.33 | 18.42 | 4.65 | 15.63 | 36.59 | 28.68 | 139.99 |
| L3 × T1 | 61.50 | 16.02 | 3.85 | 14.33 | 34.88 | 33.27 | 129.14 |
| L4 × T1 | 67.17 | 16.37 | 4.45 | 13.35 | 32.95 | 33.15 | 147.40 |
| L5 × T1 | 62.83 | 16.42 | 4.80 | 15.10 | 35.20 | 35.90 | 161.21 |
| L6 × T1 | 65.17 | 18.37 | 4.25 | 13.15 | 37.20 | 32.26 | 112.39 |
| L7 × T1 | 66.50 | 15.82 | 4.40 | 13.42 | 33.50 | 33.41 | 149.85 |
| L8 × T1 | 63.67 | 16.02 | 4.83 | 15.39 | 36.84 | 29.33 | 142.91 |
| L9 × T1 | 63.17 | 14.52 | 4.55 | 14.53 | 34.15 | 30.19 | 156.19 |
| L10 × T1 | 62.50 | 15.29 | 4.25 | 16.47 | 31.98 | 24.90 | 135.26 |
| L1 × T2 | 62.17 | 16.97 | 4.95 | 16.08 | 37.68 | 29.60 | 152.94 |
| L2 × T2 | 62.50 | 17.77 | 4.61 | 15.19 | 38.80 | 27.41 | 151.26 |
| L3 × T2 | 61.67 | 18.67 | 4.00 | 14.08 | 37.64 | 31.86 | 136.54 |
| L4 × T2 | 65.50 | 17.87 | 4.05 | 13.37 | 36.84 | 36.25 | 162.17 |
| L5 × T2 | 61.17 | 17.52 | 4.40 | 17.17 | 37.40 | 33.70 | 174.18 |
| L6 × T2 | 65.00 | 17.77 | 4.55 | 15.00 | 29.15 | 27.39 | 120.79 |
| L7 × T2 | 66.00 | 16.62 | 4.92 | 13.30 | 32.05 | 27.17 | 128.99 |
| L8 × T2 | 64.17 | 16.77 | 4.53 | 14.03 | 33.80 | 32.91 | 135.66 |
| L9 × T2 | 63.50 | 16.72 | 4.25 | 14.07 | 32.45 | 28.44 | 120.07 |
| L10 × T2 | 59.33 | 17.52 | 4.50 | 15.33 | 30.95 | 29.89 | 126.05 |
| L1 × T3 | 65.50 | 16.92 | 4.45 | 15.60 | 34.94 | 24.83 | 142.07 |
| L2 × T3 | 63.00 | 18.57 | 4.10 | 15.90 | 31.93 | 30.14 | 152.50 |
| L3 × T3 | 60.17 | 17.97 | 4.20 | 14.63 | 36.53 | 30.08 | 118.29 |
| L4 × T3 | 66.83 | 16.77 | 4.05 | 13.20 | 33.43 | 32.51 | 141.82 |
| L5 × T3 | 62.50 | 17.22 | 4.30 | 15.33 | 32.49 | 37.21 | 148.29 |
| L6 × T3 | 60.50 | 16.77 | 4.25 | 13.63 | 30.00 | 32.23 | 148.04 |
| L7 × T3 | 65.50 | 16.49 | 4.55 | 15.62 | 29.84 | 30.46 | 100.26 |
| L8 × T3 | 65.50 | 16.42 | 4.38 | 16.50 | 31.37 | 29.53 | 116.34 |
| L9 × T3 | 63.67 | 16.87 | 4.90 | 15.33 | 35.72 | 31.31 | 117.31 |
| L10 × T3 | 61.33 | 16.42 | 4.55 | 16.15 | 32.00 | 26.24 | 100.00 |
| TWC-368 | 65.50 | 17.52 | 4.60 | 15.97 | 34.45 | 31.51 | 146.00 |
| LSD $_{0.05}$ | 1.64 | 1.05 | 0.25 | 1.11 | 2.34 | 3.11 | 7.97 |
| LSD $_{0.01}$ | 2.16 | 1.38 | 0.33 | 1.46 | 3.07 | 4.09 | 10.47 |

DTS: days to 50% silking, EL: ear length, ED: ear diameter, NRPE: number of rows/ear, NKPR: number of kernels/row, HKW: hundred kernel weight and GYPP: grain yield/plant.

### 3.1.3. General (GCA) and Specific (SCA) Combining Ability Effects

High positive values of ($\hat{g}_i$) effects are important for all agronomic traits except DTS, where negative values are desirable. The GCA effects for evaluated 10 inbred lines and three testers varied widely among measured traits (Table 3). The lines L2, L3, L5, and L10 displayed the highest significant ($p < 0.05$) and negative GCA effects for DTS. On the contrary, the highest significant ($p < 0.05$) and positive GCA effects were recorded by lines L2, L3, and L6 for EL; L7, L8, and L9 for ED; L1, L2, L5, and L10 for NRPE; L1, L2, and L3 for NKPR; L4 and L5 for HKW; L1, L2, L4, and L5 for GYPP. Regarding the testers, the

highest significant and desirable GCA effects were obtained by T1 (SC-168) for NKPR and GYPP; T2 (SC-176) for DTS, EL, ED, NKPR, and GYPP and T3 (TWC-352) for only NRPE.

**Table 3.** General combining ability effects (GCA) for the evaluated 10 inbred lines and three testers for agronomic traits over two locations.

| Genotypes | DTS | EL | ED | NRPE | NKPR | HKW | GYPP |
|---|---|---|---|---|---|---|---|
| Inbred Lines | | | | | | | |
| L1 | 0.46 | −0.19 | 0.08 | 0.79 ** | 1.52 ** | −3.38 ** | 4.06 * |
| L2 | −0.87 * | 1.31 ** | 0.03 | 0.70 ** | 1.69 ** | −1.84 ** | 11.43 ** |
| L3 | −2.37 ** | 0.61 ** | −0.40 ** | −0.52 * | 2.26 ** | 1.16 | −8.49 ** |
| L4 | 3.02 ** | 0.06 | −0.24 ** | −1.57 ** | 0.32 | 3.39 ** | 13.98 ** |
| L5 | −1.32 ** | 0.11 | 0.08 | 0.99 ** | 0.95 | 5.02 ** | 24.74 ** |
| L6 | 0.07 | 0.70 ** | −0.07 | −0.95 ** | −1.97 ** | 0.04 | −9.41 ** |
| L7 | 2.52 ** | −0.63 ** | 0.20 ** | −0.76 ** | −2.29 ** | −0.23 | −10.12 ** |
| L8 | 0.96 ** | −0.54 * | 0.16 ** | 0.44 | −0.08 | 0.01 | −4.85 ** |
| L9 | −0.04 | −0.90 ** | 0.15 ** | −0.23 | 0.02 | −0.60 | −5.29 ** |
| L10 | −2.43 ** | −0.53 * | 0.01 | 1.11 ** | −2.44 ** | −3.57 ** | −16.05 ** |
| LSD (gi) $_{0.05}$ | 0.68 | 0.43 | 0.11 | 0.46 | 0.97 | 1.28 | 3.30 |
| LSD (gi) $_{0.01}$ | 0.89 | 0.57 | 0.14 | 0.60 | 1.27 | 1.68 | 4.34 |
| Testers | | | | | | | |
| T1 (SC-168) | 0.42 * | −0.58 ** | −0.01 | −0.21 | 0.67 * | 0.24 | 3.61 ** |
| T2 (SC-176) | −0.38 * | 0.48 ** | 0.06 * | −0.11 | 0.59 * | −0.12 | 4.38 ** |
| T3 (TWC-352) | −0.03 | 0.10 | −0.05 | 0.32 * | −1.26 ** | −0.12 | −7.99 ** |
| LSD (gi) $_{0.05}$ | 0.37 | 0.24 | 0.06 | 0.25 | 0.53 | 0.70 | 1.81 |
| LSD (gi) $_{0.01}$ | 0.49 | 0.31 | 0.08 | 0.33 | 0.70 | 0.92 | 2.38 |

\* and \*\* indicate *p*-value < 0.05 and 0.01, respectively, DTS: days to 50% silking, EL: ear length, ED: ear diameter, NRPE: number of rows/ear, NKPR: number of kernels/row, HKW: hundred kernel weight and GYPP: grain yield/plant.

The SCA effects in Table 4 revealed that the highest desirable significant (*p* < 0.05) and negative effects for DTS were recorded by four crosses L8 × T1, L1 × T2, L10 × T2, and L6 × T3. In contrast, the highest positive and significant (*p* < 0.05) SCA effects were obtained by L2 × T1 and L6 × T1 for EL. Additionally, the hybrids L2 × T1, L4 × T1, L5 × T1, L8 × T1, L1 × T2, L7 × T2, and L9 × T3 exhibited the uppermost positive SCA effects for ED. The highest positive effects for NRPE were obtained by hybrids L5 × T2, L6 × T2, L7 × T3, and L8 × T3. Likewise, hybrids L6 × T1, L8 × T1, L2 × T2, L4 × T2, L5 × T2, and L9 × T3 had the highest positive SCA effects for NKPR; L7 × T1, L1 × T2, L4 × T2, L8 × T2, and L10 × T2 for HKW and L7 × T1, L8 × T1, L9 × T1, L10 × T1, L1 × T2, L4 × T2, L5 × T2, L1 × T3, L2 × T3, and L6 × T3 for GYPP.

Notably, there were no specific cross combinations that exhibited favorable SCA effects for all evaluated characteristics. Nevertheless, some hybrids had beneficial effects for GYPP as well as recorded desirable SCA effects for one or more of its components. For instance, the hybrid L4 × T2 had desirable SCA effects for NKPR, HKW, and GYPP and also the cross L5 × T2 had desirable SCA effects for NRPE, NKPR, and GYPP.

*3.2. Late Wilt Trial*

3.2.1. ANOVA and Line × Tester Analysis for Late Wilt Resistance

The analysis of variance for resistance to LWD revealed significant differences (*p* < 0.05) among genotypes (G), crosses (C), lines (L), testers (T), and their interactions with nitrogen levels except for testers under high nitrogen level and T × N interaction which were not significant (Table 5). Moreover, it is worth noting that the differences between the two nitrogen levels were not significant (*p* > 0.05). The SCA variance was higher than GCA variance under both nitrogen levels. Additionally, the magnitude of SCA × N interaction was higher than those of GCA × N interaction (Table 5).

**Table 4.** Specific combining ability (SCA) effects of thirty test-crosses for all measured traits over two locations.

| Hybrid | DTS | EL | ED | NRPE | NKPR | HKW | GYPP |
|---|---|---|---|---|---|---|---|
| L1 × T1 | −0.19 | 0.20 | −0.39 ** | −0.16 | −2.08 * | −0.28 | −17.54 ** |
| L2 × T1 | −0.69 | 0.75 * | 0.20 * | 0.27 | 0.15 | −0.31 | −11.54 ** |
| L3 × T1 | −0.03 | −0.95 * | −0.16 | 0.19 | −2.14 * | 1.29 | −2.46 |
| L4 × T1 | 0.25 | −0.05 | 0.27 ** | 0.25 | −2.12 * | −1.06 | −6.67 * |
| L5 × T1 | 0.25 | −0.05 | 0.31 ** | −0.56 | −0.50 | 0.05 | −3.63 |
| L6 × T1 | 1.19 * | 1.31 ** | −0.09 | −0.57 | 4.42 ** | 1.39 | −18.29 ** |
| L7 × T1 | 0.08 | 0.09 | −0.21* | −0.49 | 1.04 | 2.82 * | 19.88 ** |
| L8 × T1 | −1.19 * | 0.20 | 0.26 ** | 0.29 | 2.17 * | −1.50 | 7.66 ** |
| L9 × T1 | −0.69 | −0.94 * | −0.01 | 0.09 | −0.62 | −0.04 | 21.39 ** |
| L10 × T1 | 1.03 | −0.54 | −0.18 | 0.69 | −0.33 | −2.36 * | 11.21 ** |
| L1 × T2 | −1.39 * | −0.26 | 0.39 ** | 0.53 | 1.48 | 2.52 * | 8.02 ** |
| L2 × T2 | 0.27 | −0.96 * | 0.10 | −0.28 | 2.43 ** | −1.21 | −1.04 |
| L3 × T2 | 0.94 | 0.64 | −0.07 | −0.16 | 0.70 | 0.24 | 4.17 |
| L4 × T2 | −0.62 | 0.39 | −0.19 * | 0.17 | 1.84 * | 2.40 * | 7.32 * |
| L5 × T2 | −0.62 | −0.01 | −0.16 | 1.41 ** | 1.78 * | −1.78 | 8.57 ** |
| L6 × T2 | 1.83 ** | −0.35 | 0.14 | 1.18 ** | −3.56 ** | −3.12 ** | −10.67 ** |
| L7 × T2 | 0.38 | −0.17 | 0.24 * | −0.70 | −0.34 | −3.06 ** | −1.76 |
| L8 × T2 | 0.11 | −0.11 | −0.11 | −1.16 ** | −0.79 | 2.44 * | −0.35 |
| L9 × T2 | 0.44 | 0.21 | −0.37 ** | −0.47 | −2.25 ** | −1.42 | −15.50 ** |
| L10 × T2 | −1.34 * | 0.63 | 0.01 | −0.54 | −1.29 | 3.00 ** | 1.23 |
| L1 × T3 | 1.59 ** | 0.07 | −0.002 | −0.38 | 0.60 | −2.24 * | 9.52 ** |
| L2 × T3 | 0.42 | 0.22 | −0.31 ** | 0.01 | −2.59 ** | 1.52 | 12.58 ** |
| L3 × T3 | −0.91 | 0.32 | 0.23* | −0.03 | 1.44 | −1.53 | −1.70 |
| L4 × T3 | 0.37 | −0.33 | −0.09 | −0.42 | 0.28 | −1.34 | −0.65 |
| L5 × T3 | 0.37 | 0.07 | −0.15 | −0.85 * | −1.28 | 1.74 | −4.94 |
| L6 × T3 | −3.02 ** | −0.97 * | −0.05 | −0.61 | −0.86 | 1.73 | 28.96 ** |
| L7 × T3 | −0.47 | 0.08 | −0.02 | 1.19 ** | −0.70 | 0.24 | −18.12 ** |
| L8 × T3 | 1.09 | −0.08 | −0.15 | 0.87 * | −1.38 | −0.94 | −7.31 * |
| L9 × T3 | 0.26 | 0.73 | 0.38 ** | 0.37 | 2.87 ** | 1.46 | −5.89 * |
| L10 × T3 | 0.31 | −0.09 | 0.16 | −0.15 | 1.61 | −0.64 | −12.44 ** |
| LSD Sij $_{0.05}$ | 1.17 | 0.75 | 0.19 | 0.8 | 1.68 | 2.21 | 5.72 |
| LSD Sij $_{0.01}$ | 1.54 | 0.98 | 0.25 | 1.05 | 2.20 | 2.91 | 7.52 |

* and ** indicate *p*-value < 0.05 and 0.01, respectively. DTS: days to 50% silking, EL: ear length, ED: ear diameter, NRPE: number of rows/ear, NKPR: number of kernels/row, HKW: hundred kernel weight and GYPP: grain yield/plant.

### 3.2.2. Mean Performance

The performance of 30 test-crosses and check hybrid (TWC-368) for resistance to LWD are presented in Table 6. The resistance ranged from 67.41 to 100% (with an average of 95.04%) under low nitrogen level, from 79.40 to 100% (with an average of 95.18%) under high nitrogen level and 80.58 to 100% (with an average of 95.11) under both nitrogen levels. Out of 30 test-crosses 18 hybrids showed high resistance (>95%) under low N and high N nitrogen levels. In addition, four crosses L2 × T2, L4 × T2, L6 × T2, and L8 × T3 exhibited 100% resistance to LWD under both N levels. Furthermore, interestingly, the crosses L5 × T1, L9 × T1, L4 × T2, and L5 × T2 that displayed high grain yield also had high resistance to LWD.

### 3.2.3. General (GCA) and Specific (SCA) Combining Ability Effects

The GCA effects for evaluated 10 inbred lines and three testers varied in resistance to LWD trait (Table 7). High positive values of GCA effects are important for resistance to LWD. Over nitrogen levels, the lines L4, L5, L6, L7, and L8 exhibited the highest significant (*p* < 0.05) and positive GCA effects. Regarding the testers, the highest significant and desirable GCA effects were obtained by T2 (SC-176) for resistance to LWD. Furthermore, the SCA effects in Table 8 indicated that the highest significant and positive SCA effects for resistance to LWD were assigned for the crosses L2 × T2, L1 × T3, and L8 × T3 over the tested environments.

**Table 5.** Ordinary analysis of variance and line × tester analysis for resistance to LWD under two nitrogen levels separately (Separ.) and combined data (Comb.) for late wilt resistance.

| Source of Variance | Nitrogen Levels Separately | | | Combined | |
| --- | --- | --- | --- | --- | --- |
| | DF | Low N | High N | DF | Mean Squares |
| | Analysis of variance Mean squares | | | | |
| Nitrogen (N) | - | - | - | 1 | 13.98 |
| Replication (Nitrogen) | - | - | - | 4 | 102.55 |
| Genotypes (G) | 30 | 205.67 ** | 193.55 ** | 30 | 291.55 ** |
| Crosses (C) | 29 | 212.76 ** | 192.94 ** | 29 | 297.72 ** |
| C Vs. Checks | 1 | 0.26 | 210.41 ** | 1 | 112.67 |
| Genotype × N | - | - | - | 30 | 107.67 ** |
| Crosses × N | - | - | - | 29 | 108.00 ** |
| C Vs. Checks × N | - | - | - | 1 | 98.00 |
| Error | 60 | 39.35 | 48.30 | 120 | 43.82 |
| | Line × tester analysis | | | | |
| Lines (L) | 9 | 368.51 ** | 415.06 ** | 9 | 667.17 ** |
| Testers (T) | 2 | 147.25 * | 89.34 | 2 | 201.25 * |
| L × T | 18 | 142.15 ** | 93.44 ** | 18 | 123.72 ** |
| L × N | - | - | - | 9 | 116.40 ** |
| T × N | - | - | - | 2 | 35.34 |
| L × T × N | - | - | - | 18 | 111.87 ** |
| Error | 58 | 39.64 | 49.92 | 116 | 44.78 |
| $K^2$ GCA | | 1.32 | 1.86 | | 9.99 |
| $K^2$ SCA | | 34.17 | 14.51 | | 13.16 |
| $K^2$ GCA × N | | - | - | | 1.59 |
| $K^2$ SCA × N | | - | - | | 22.37 |

* and ** indicate *p*-value < 0.05 and 0.01, respectively, DF: degree of freedom.

### 3.3. Interrelationship Among Measured Traits

Principal components were calculated to visualize the relationship among the measured traits. The first two principal components reflected most of the variance which was 71.83% (61.89% and 9.94% by PC1 and PC2, respectively). Therefore, the two PCs were utilized to perform the PC-biplot (Figure 2). The acute angles among trait vectors reveal a robust positive association, while, angles more than 90° suggest no to negative association among traits. Accordingly, a strong positive association was demonstrated between grain yield/plant and each of hundred kernel weight, number of kernels/row and ear length. Moreover, a strong positive association was detected between resistance to LWD under low and high N levels, days to 50% silking and ear diameter. On the contrary, the number of rows/ear displayed a negative association with all traits.

### 3.4. Genetic Analysis Based on Microsatellites

Out of the analyzed 24 SSR primers, 13 polymorphic markers were detected throughout maize genome. The polymorphic SSR markers were applied to determine the genetic diversity among the investigated genotypes in this study. It generated a total of 54 reproducible DNA bands/alleles. The allele numbers per locus ranged from 3 (phi308707, phi024 and phi301654) to 6 (umc1225), with an average of 4.15 alleles/locus (Table 9). The amplified fragments generated by the highest polymorphic markers; umc1033 and umc1225 are shown in Figure 3. The major frequent allele frequency (0.46) was observed for the markers phi233376, phi024, umc2038, phi453121, and phi96100, while the lowest (0.33) was observed for umc1152 with an average of 0.42. The genetic diversity varied from 0.57 (umc1152) to 0.75 (umc1033 and umc 1225) with an average of 0.69. Furthermore, the average of polymorphic information content (PIC) was 0.64 with a range of 0.57 (phi308707 and phi024) to 0.72 (umc 1225).

Pairwise genetic distances (GD) between the inbred lines and testers based on SSR markers varied from 0.60 (between L8 and T3) to 0.97 (between L1 and T2), with an average

of 0.81 (Figure 4A). The GD means of the 10 inbreds with T1, T2, and T3 testers were 0.76, 0.85, and 0.82, respectively (Figure 4B). The neighbor-joining tree on the basis of genetic distance matrix divided the 13 genotypes (10 inbred lines and three testers) into three major clusters with internal sub-clusters showing varying degrees of diversity (Figure 5). Cluster I consisted of the inbreds L1, L2, and L3. Meanwhile, cluster II included the inbreds L5, L6, L7, and L4. Moreover, cluster III contained the inbreds L8, L9, and L10, as well as the three testers T1, T2, and T3. This cluster comprised of two sub-clusters; the first cluster involved four genotypes, L8, L9, L10, and T1, while the second sub-cluster consisted of T2 and T3 testers.

**Table 6.** Mean performance of the 30 hybrids and the check hybrid TW-368 for resistance to late wilt (%) under two nitrogen levels and their combined data.

| Hybrid | Low N | High N | Combined |
|---|---|---|---|
| L1 × T1 | 95.00 | 94.44 | 94.72 |
| L2 × T1 | 97.50 | 91.18 | 94.34 |
| L3 × T1 | 93.45 | 79.40 | 86.43 |
| L4 × T1 | 100.00 | 97.37 | 98.68 |
| L5 × T1 | 97.37 | 100.00 | 98.68 |
| L6 × T1 | 100.00 | 97.92 | 98.96 |
| L7 × T1 | 100.00 | 98.33 | 99.17 |
| L8 × T1 | 92.71 | 100.00 | 96.35 |
| L9 × T1 | 97.50 | 100.00 | 98.75 |
| L10 × T1 | 90.54 | 80.00 | 85.27 |
| L1 × T2 | 94.64 | 94.12 | 94.38 |
| L2 × T2 | 100.00 | 100.00 | 100.00 |
| L3 × T2 | 92.88 | 94.68 | 93.78 |
| L4 × T2 | 100.00 | 100.00 | 100.00 |
| L5 × T2 | 97.73 | 100.00 | 98.86 |
| L6 × T2 | 100.00 | 100.00 | 100.00 |
| L7 × T2 | 100.00 | 97.50 | 98.75 |
| L8 × T2 | 97.50 | 100.00 | 98.75 |
| L9 × T2 | 87.50 | 95.65 | 91.58 |
| L10 × T2 | 92.50 | 83.92 | 88.21 |
| L1 × T3 | 96.15 | 100.00 | 98.08 |
| L2 × T3 | 82.62 | 94.74 | 88.68 |
| L3 × T3 | 89.73 | 82.77 | 86.25 |
| L4 × T3 | 95.45 | 94.64 | 95.05 |
| L5 × T3 | 97.22 | 96.88 | 97.05 |
| L6 × T3 | 95.83 | 96.15 | 95.99 |
| L7 × T3 | 100.00 | 93.75 | 96.88 |
| L8 × T3 | 100.00 | 100.00 | 100.00 |
| L9 × T3 | 100.00 | 98.08 | 99.04 |
| L10 × T3 | 67.41 | 93.75 | 80.58 |
| TWC-368 | 96.67 | 100.00 | 98.33 |
| LSD $_{0.05}$ | 10.14 | 11.24 | 7.49 |
| LSD $_{0.01}$ | 13.40 | 14.85 | 9.84 |

**Table 7.** General combining ability effects (GCA) for the evaluated 10 inbred lines and three testers for resistance to late wilt under two nitrogen levels and their combined data.

| Genotypes | Low N | High N | Combined |
|---|---|---|---|
| Inbred Lines | | | |
| L1 | −1.34 | 1.05 | −0.15 |
| L2 | −1.47 | 0.26 | −0.61 |
| L3 | −7.38 ** | −13.06 ** | −10.22 ** |
| L4 | 5.49 ** | 2.39 | 3.94 * |
| L5 | 1.37 | 5.77 * | 3.57 * |
| L6 | 5.62 ** | 3.24 | 4.43 ** |
| L7 | 8.80 ** | −0.75 | 4.02 * |
| L8 | 1.13 | 8.51 ** | 4.82 ** |
| L9 | 0.72 | 3.13 | 1.92 |
| L10 | −12.94 ** | −10.53 ** | −11.73 ** |
| LSD (gi) $_{0.05}$ | 4.11 | 4.62 | 3.09 |
| LSD (gi) $_{0.01}$ | 5.41 | 6.07 | 4.06 |
| Testers | | | |
| T1 (SC-168) | 0.74 | −0.90 | −0.08 |
| T2 (SC-176) | 1.75 | 1.99 | 1.87 * |
| T3 (TWC-352) | −2.49 * | −1.09 | −1.79 * |
| LSD (gi) $_{0.05}$ | 2.25 | 2.53 | 1.69 |
| LSD (gi) $_{0.01}$ | 2.96 | 3.32 | 2.23 |

* and ** indicate *p*-value < 0.05 and 0.01, respectively.

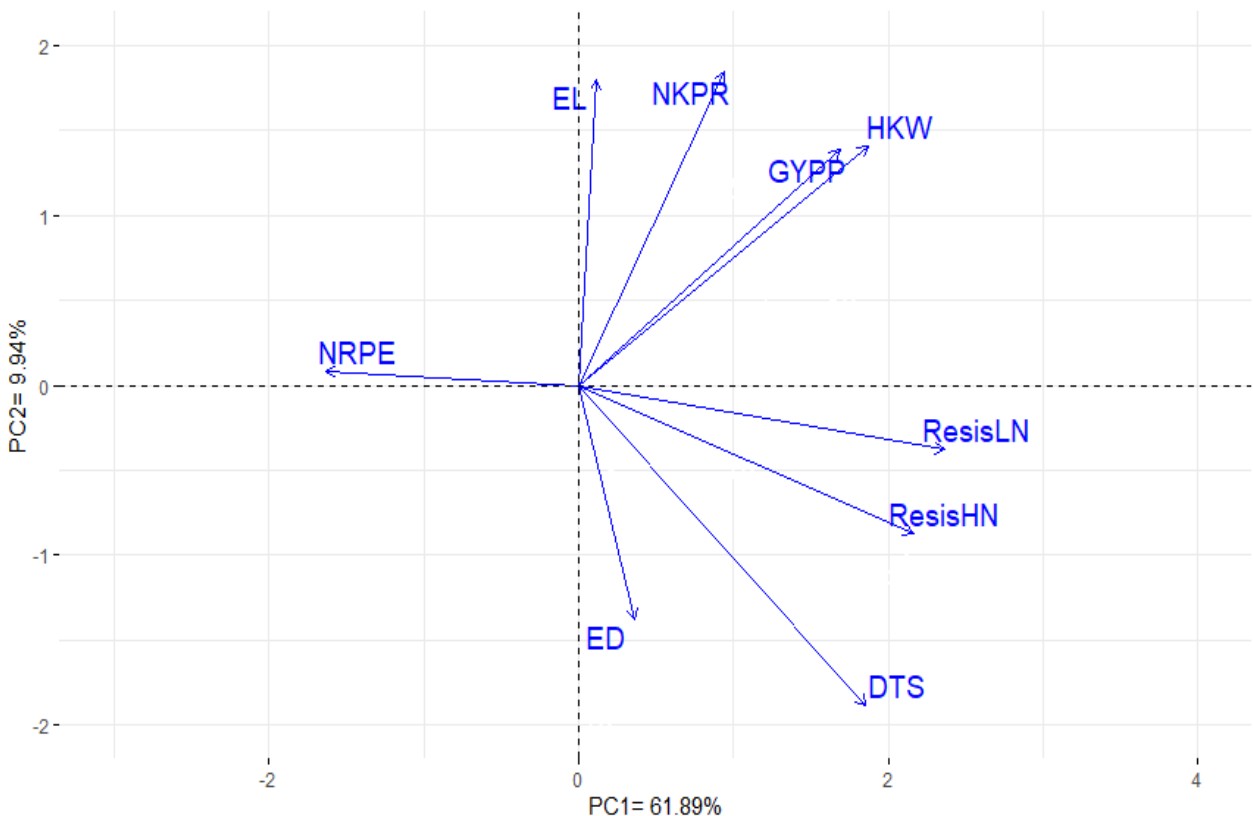

**Figure 2.** PC-biplot for measured traits of the evaluated thirty test-crosses. DTS: days to 50% silking, EL: ear length, ED: ear diameter, NRPE: number of rows/ear, NKPR: number of kernels/row, HKW: hundred kernel weight, GYPP: grain yield/plant, ResisLN: resistance to late wilt disease under low N level, ResisHN: resistance to late wilt disease under high N level.

**Table 8.** Specific combining ability (SCA) effects of thirty test-crosses for resistance to late wilt under two nitrogen levels and their combined data.

| Hybrid | Low N | High N | Combined |
|---|---|---|---|
| L1 × T1 | −1.04 | −2.67 | −1.85 |
| L2 × T1 | 2.20 | −4.88 | −1.34 |
| L3 × T1 | 0.84 | −4.47 | −1.81 |
| L4 × T1 | 2.57 | −0.51 | 1.03 |
| L5 × T1 | −0.85 | 3.64 | 1.40 |
| L6 × T1 | 2.44 | −0.51 | 0.96 |
| L7 × T1 | −0.74 | 4.19 | 1.73 |
| L8 × T1 | −8.74 * | 0.90 | −3.92 |
| L9 × T1 | 0.01 | 6.29 | 3.15 |
| L10 × T1 | 3.30 | −1.98 | 0.66 |
| L1 × T2 | −2.45 | −5.89 | −4.17 |
| L2 × T2 | 8.51 * | 6.27 | 7.39 ** |
| L3 × T2 | −1.04 | 6.26 | 2.61 |
| L4 × T2 | 1.56 | 4.13 | 2.84 |
| L5 × T2 | −1.33 | 0.75 | −0.29 |
| L6 × T2 | 1.42 | 3.29 | 2.36 |
| L7 × T2 | −1.75 | −0.06 | −0.91 |
| L8 × T2 | −1.42 | −1.99 | −1.70 |
| L9 × T2 | −10.58 ** | −6.33 | −8.45 ** |
| L10 × T2 | 7.08 | −6.42 | 0.33 |
| L1 × T3 | 3.49 | 8.55 * | 6.02 * |
| L2 × T3 | −10.71 ** | −1.39 | −6.05 * |
| L3 × T3 | 0.20 | −1.79 | −0.79 |
| L4 × T3 | −4.13 | −3.62 | −3.88 |
| L5 × T3 | 2.18 | −4.39 | −1.11 |
| L6 × T3 | −3.87 | −2.77 | −3.32 |
| L7 × T3 | 2.49 | −4.13 | −0.82 |
| L8 × T3 | 10.16 ** | 1.09 | 5.62 * |
| L9 × T3 | 10.57 ** | 0.05 | 5.31 |
| L10 × T3 | −10.38 ** | 8.40 * | −0.99 |
| LSD Sij $_{0.05}$ | 7.12 | 8.00 | 5.35 |
| LSD Sij $_{0.01}$ | 9.36 | 10.51 | 7.04 |

\* and \*\* indicate *p*-value < 0.05 and 0.01, respectively.

**Table 9.** Statistics of the thirteen SSR markers applied in this study.

| Marker | Chromosome Number | No. of Alleles | Major Allele Frequency | Gene Diversity | PIC |
|---|---|---|---|---|---|
| phi308707 | 1 | 3 | 0.42 | 0.65 | 0.57 |
| phi96100 | 2 | 4 | 0.46 | 0.69 | 0.64 |
| phi453121 | 3 | 4 | 0.46 | 0.70 | 0.67 |
| umc1963 | 4 | 4 | 0.38 | 0.70 | 0.64 |
| umc2038 | 4 | 4 | 0.46 | 0.69 | 0.64 |
| phi024 | 5 | 3 | 0.46 | 0.64 | 0.57 |
| umc1225 | 5 | 6 | 0.42 | 0.75 | 0.72 |
| umc1014 | 6 | 5 | 0.41 | 0.70 | 0.65 |
| umc2332 | 7 | 4 | 0.38 | 0.71 | 0.66 |
| phi233376 | 8 | 5 | 0.46 | 0.71 | 0.67 |
| umc1033 | 9 | 5 | 0.38 | 0.75 | 0.71 |
| umc1152 | 10 | 4 | 0.33 | 0.57 | 0.61 |
| phi301654 | 10 | 3 | 0.38 | 0.65 | 0.58 |
| Mean | | 4.15 | 0.42 | 0.69 | 0.64 |

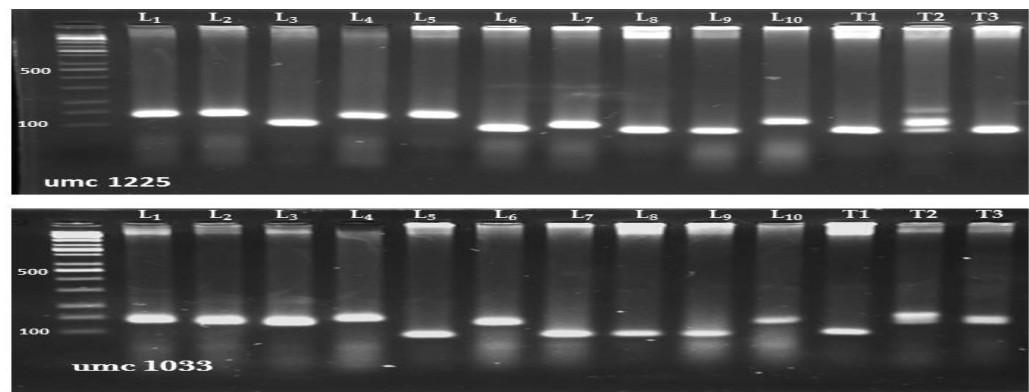

**Figure 3.** PCR amplified fragments for highly polymorphic SSR markers; umc1033 and umc1225. M is a 100 bp DNA ladder.

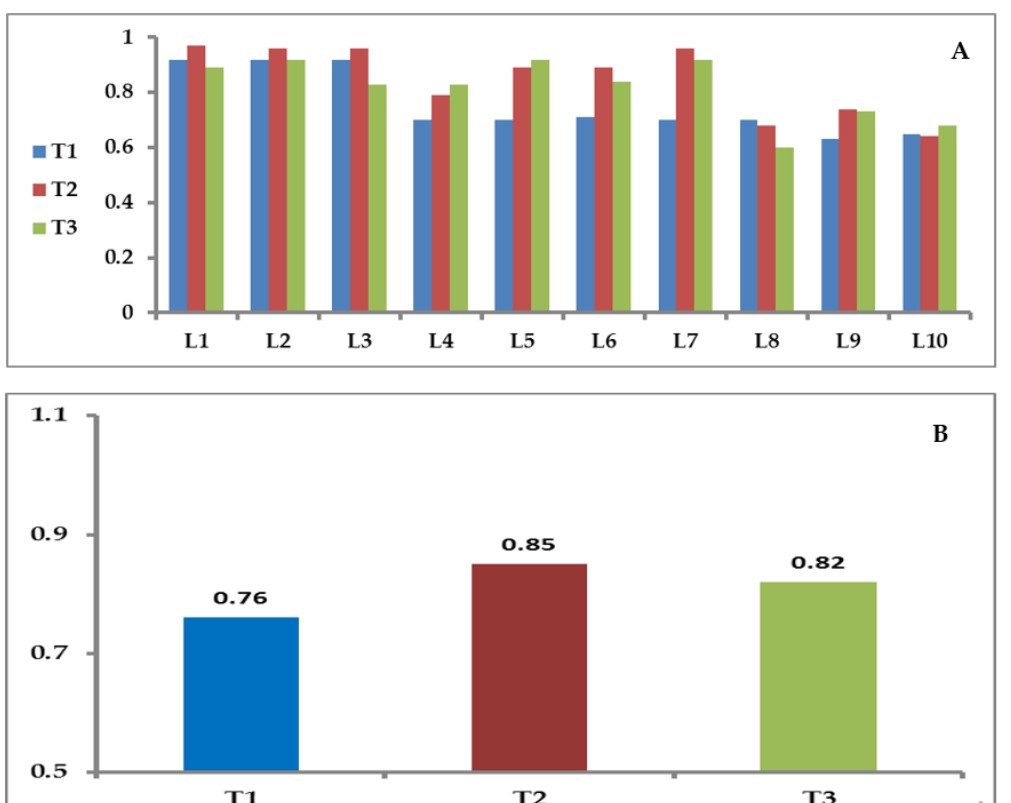

**Figure 4.** (**A**) Pairwise genetic distance between the inbred lines and testers based on SSR markers; T1: SC-168, T2: SC-176, and T3: TWC-352. (**B**) Genetic distance (GD) means of the 10 inbred lines with each tester based on SSR markers, T1 (SC-168), T2 (SC-176), and T3 (TWC-352).

### 3.5. Correlation between Parental GD, Hybrid Performance and SCA

The relationship among parental GD and hybrid performance as well as SCA effects was tested to investigate the possibility of predicting hybrid performance and SCA effects in maize based on parental GD using molecular markers. The correlations between SSR marker-based GD and hybrid performance for GYPP and LWD were not significant (r = 0.15 and r = 0.05, respectively). Similarly, GD had no significant correlation with SCA effects for GYPP (r = −0.08) and LWD (r = −0.15). Conversely, SCA effects of the evaluated crosses showed a strong significant ($p < 0.05$) correlation with the performance of $F_1$ hybrid for both GYPP, (r = 0.65 *) and LWD (r = 0.43 *).

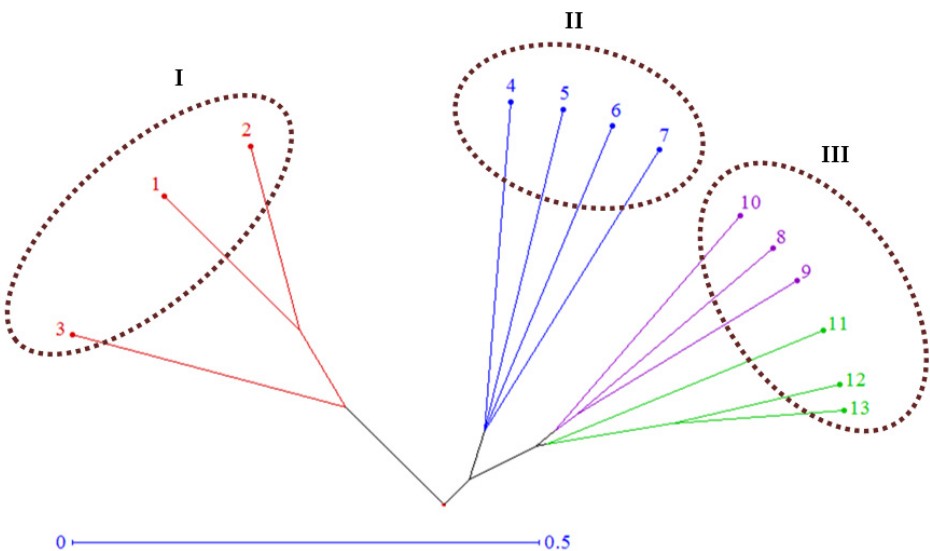

**Figure 5.** Neighbor-joining tree of the 13 genotypes (10 lines and 3 testers) based on SSR analysis. The inbred lines are 1–10 and testers are 11–13, in the same order as Table 3.

## 4. Discussion

The present study employed 10 diverse yellow maize inbred lines assembled from different origins and three high-yielding testers were crossed to obtain 30 test-crosses. The obtained test-crosses plus the check hybrid were evaluated in field trials across two different locations as well as for resistance to LWD under two nitrogen levels. The combined ANOVA for GYPP and its related agronomic traits as well as for resistance to LWD depicted highly significant differences among lines, testers, and their interaction. The detected genetic divergences reflected the potential of used inbred lines, testers, and their test-cross progenies in ameliorating the genetic diversity that could be exploited through maize breeding for alleviating grain yield and resistance to LWD. The genetic variability facilitates selecting favorable alleles and thereby promising hybrids based on agronomic performance and resistance to LWD. Similarly, genetic differences were previously reported among maize hybrids for grain yield by Elmyhun et al. [20], Amegbor et al. [41], Ajala et al. [42], and Kamara et al. [43], as well as for resistance to LWD by El-Hosary and El-Fiki [28], Biradar [44], Biradar et al. [45], and Mosa et al. [46]. The significant variance of tested locations for measured agronomic traits indicates their difference. Variation in soil and climatic conditions between the two locations could be the main cause of the observed significant variance. Similar observations were reported by other studies, such as Ajala et al. [42] and Badu-Apraku et al. [47]. In contrast, the difference between the two nitrogen levels was not significant for resistance to LWD suggesting that the resistance to LWD was not affected by nitrogen levels. This finding concurs with Mosa et al. [46], who elucidated the expression of resistance to LWD was consistent under different nitrogen levels. In this context, Mansour et al. [48], depicted that the agricultural practices tend to balance N fertilization for ecological and economic reasons. Moreover, Singh and Siradhana [49] manifested that balanced nitrogen fertilization can diminish disease severity, although it does not provide complete control. On the other hand, Ortiz-Bustos et al. [50] disclosed that the development of LWD symptoms was impacted by air temperature, air humidity, and water deficit, notwithstanding it is principally controlled by genetic resistance.

Investigating the interrelationships among plant traits can provide valuable information. The biplot of principal components is a statistical procedure for assessing the interrelationships among evaluated traits. In the present study, a strong positive association was reflected between resistance to LWD under low and high N levels. Moreover, a positive association was detected between resistance to LWD and grain yield per plant. Various re-

searchers have applied PC-biplot to understand the interrelationships among studied traits as Desoky et al. [51], Mansour et al. [52], Moustafa et al. [53], and El-Sanatawy et al. [54].

One of the major aims of the current study was to identify high-yielding and LWD resistant hybrids. Four crosses, L5 × T1, L9 × T1, L4 × T2, and L5 × T2, displayed high grain yield as well as high resistance to LWD disease. However, the interaction between the evaluated hybrids and tested locations was significant for all agronomic traits. Accordingly, the superior hybrids should be further tested extensively across multi-environment trials to emphasize the consistency of their performance before commercialization. The obtained results are consistent with previous findings of Beyene et al. [19], Abd El-Aty et al. [55], Badu-Apraku and Oyekunle [56], and Badu-Apraku et al. [57].

The variance due to SCA demonstrated a higher magnitude than GCA for all measured traits. This finding suggested that the non-additive gene effects had more important role in regulating the inheritance of all assessed traits including GYPP and LWD. Accordingly, the hybridization method is useful in improving these characteristics by exploiting the hybrid vigor. Similar results were obtained by Badu-Apraku et al. [30], Kamara et al. [43], Mosa et al. [46], Makumbi et al. [58], and Derera et al. [59]. They demonstrated that non-additive effects regulated the inheritance of maize grain yield and resistance to LWD. On the contrary, other studies manifested that the additive genetic action mainly controlled the expression of these traits as Badu-Apraku [18], El-Hosary and El-Fiki [28], Badu-Apraku et al. [60], Annor et al. [61], and Oyetunde et al. [62]. The preponderance of SCA × location interaction over GCA × location for all measured traits implied that the non-additive genetic effects were more influenced by the environment than the additive ones. This result is in harmony with the findings of various researchers such as Kamara et al. [2], Abd El-Aty et al. [55], and Kamara et al. [43].

Developing maize hybrids with high yield and resistance to LWD depends principally on the utilized parents. The GCA is an important indicator for line performance in cross combinations and the potential for generating superior hybrids [32,63]. In the current study, significant and desirable GCA effects for the measured traits were demonstrated among studied inbred lines and testers. The inbreds L2, L3, L5, and L10 were recognized as good combiners for earliness. Furthermore, lines L1, L2, L4, and L5 were the superior general combiners for GYPP and certain components indicating their ability to increase grain yield in cross combinations. Besides, the lines L4, L5, L6, L7, and L8 were identified as excellent combiners for resistance to LWD. Subsequently, exploiting these inbreds in breeding programs could produce progenies with enhanced resistance to LWD. Interestingly, the lines L4 and L5 reflected desirable GCA effects for GYPP and were also excellent combiners for resistance to LWD. Thus, these inbreds could transfer these beneficial alleles to their offspring for developing high-yielding and resistant hybrids to LWD. The appropriate tester should provide a precise ranking of the tested inbred lines and optimize the genetic gain [61]. However, it is difficult to find testers with all required characteristics [64]. In the current study, T2 (SC-176) was identified as a suitable tester for improving DTS, EL, ED, NKPR, GYPP, and resistance to LWD. This confirms its potential role as a donor for high yield and resistance to LWD in future maize breeding programs. The advantage of single crosses as good testers was previously deduced by Abd El-Aty et al. [55] and Kamara et al. [43].

Based on the SCA estimation, 10 hybrid combinations are considered promising specific combiners for breeding high-yielding hybrids. Out of the identified crosses, three hybrids, L9 × T1, L4 × T2, and L5 × T2, exhibited desirable SCA which coupled with high grain yield. This finding indicates a clear correspondence between positive SCA effects and high grain yield. The association between desired SCA effects and high yield performance was also disclosed by Kamara et al. [2] and Elmyhun et al. [20]. Moreover, the crosses L2 × T2, L1 × T3, L8 × T3, and L9 × T3 were the best specific combiners for improving resistance to LWD. Most of these hybrids were formed from good × good general combiners, which indicates increasing the concentration of favorable alleles. In this context, Kamara et al. [2] reported that the inclusion of at least one good general combiner

is crucial for providing good specific combinations. The hybrids with positive SCA effects for both GYPP and resistance to LWD are highly favorable in maize breeding programs. Interestingly, the hybrid L1 × T3 had desirable SCA for GYPP as well as showing high SCA for LWD resistance. This hybrid is recommended for inclusion in future breeding programs to increase maize grain yield and resistance to LWD.

The SSR markers used in the present study demonstrated the degree of genetic diversity among investigated maize inbred lines and testers. The obtained results revealed the number of alleles per locus ranged from 3 to 6 with an average of 4.15 alleles per locus which is analogous to the reported findings of Xu et al. [65] and Wegary et al. [66], which were 4.4 and 4.2 allele/locus, respectively. However, this was lower than the reported averages by Reif et al. [67] and Oppong et al. [68], which were 7.7 and 6.21 alleles/locus, respectively. The difference in mean number of alleles between studies could be attributed to using different number of markers and genotypes. The most frequent allele had an average of 0.42 indicating that 42.0% of the investigated inbreds shared a common allele at any of the tested loci. The PIC measures allelic diversity at a locus; it was high and presented 0.64 which reflected good discriminatory power of the used markers [69]. This value was lower than the described values by Aci et al. [69] and Adu et al. [70], while it was slightly higher than those reported by Badu-Apraku et al. [57], Akinwale et al. [71], and Wende et al. [72]. Moreover, the markers umc1033 and umc1225 exhibited higher discriminatory power to distinguish genotypes in this study due to their high PIC values which were 0.71 and 0.72, respectively.

Pairwise genetic distance (GD) between the evaluated 10 inbred lines and three testers was high (0.81) suggesting presence of substantial genetic diversity based on the microsatellite markers analysis. The inbred lines displayed higher average of GD with T2 tester than the other testers indicating a wide diversity between this tester and the evaluated inbreds. The SSR markers clearly grouped the parental genotypes into three main clusters. Three Egyptian lines (L1–L3) were presented in the first group while introduced lines from CIMMYT were presented in two diverse groups (second and third groups). Thereupon, crossing inbreds from diverse groups may result in more effective hybrids. It is noteworthy that the two lines L4 and L5 in cluster II gave higher grain yields in combination with T2 tester located in cluster III. This confirms the possibility of obtaining superior hybrids by crossing genotypes from different groups. This result is consistent with studies of Ajala et al. [42] and Annor et al. [73]. The obtained information from this cluster analysis could help in minimizing the number of crosses to be generated and tested in the field. The non-significant correlation recorded between SSR-based GD, hybrid performance, and SCA effects could be impacted by not using a large number of markers. However, similar results were disclosed by Menkir et al. [74], Dhliwayo et al. [75], Badu-Apraku et al. [57], and Kamara et al. [2]. Conversely, Betran et al. [34], Balestre et al. [33], Phumichai et al. [35], and Singh [76] demonstrated a significant relationship between GD and hybrid performance. Furthermore, the results demonstrated that SCA effects were significantly associated with test-cross performance for GYPP and resistance to LWD. This result is in accordance with the finding of Kamara et al. [2], Badu-Apraku et al. [57], and Mageto et al. [77], who depicted that SCA effects could be used as a good predictor for the performance of hybrids.

## 5. Conclusions

The current study displayed considerable genetic variation amongst lines, testers, and their corresponding hybrids for all evaluated traits. The non-additive genetic effects were predominantly for grain yield and resistance to LWD, which permit improving these traits through crossing and exploiting hybrid vigor. The inbred lines L2, L3, L5, and L10 were identified as excellent combiners for developing early maturity hybrids. However, the inbreds L4 and L5 are recommended for future breeding programs for increasing grain yield and resistance to LWD. The hybrids L9 × T1, L4 × T2, and L5 × T2 combined significant SCA with high grain yield. These hybrids will be further evaluated for possible releasing and cultivation commercially. The hybrid L1 × T3 exhibited excellent SCA effects for both

grain yield and resistance to LWD. Cluster analysis classified the parental genotypes into three main groups, which could help in reducing the number of crosses to be generated and evaluated in the field. SCA displayed a positive association with grain yield and resistance to LWD which could be exploited to predict the performance of hybrids.

**Supplementary Materials:** The following are available online at https://www.mdpi.com/article/10.3390/agronomy11050898/s1. Table S1. Code, name, pedigree and source of the ten parental maize inbred lines. Table S2. Physical and chemical soil characteristics of the two experimental sites. Table S3. List of SSR primers and their sequences used in the present study.

**Author Contributions:** Conceptualization, M.M.K., N.A.G., E.M., and K.M.I.; methodology M.M.K., N.A.G., E.M., and K.M.I.; software, M.M.K., E.M., A.M.S.K., and M.M.E.; validation M.M.K., N.A.G., and K.M.I.; formal analysis, M.M.K., E.M., A.M.S.K., and M.M.E.; investigation M.M.K., N.A.G., E.M., and K.M.I.; writing—original draft preparation, M.M.K., M.M.E., E.M., N.A.G., and K.M.I.; M.M.K., M.M.E., E.M., and N.A.G. and K.M.I. writing—review and editing, M.M.K., M.M.E., E.M., N.A.G., and K.M.I.; M.M.K., M.M.E., E.M., N.A.G., A.M.S.K., and K.M.I.; visualization M.M.K., M.M.E., E.M., N.A.G., and K.M.I. All authors have read and agreed to the published version of the manuscript.

**Funding:** This research received no external funding.

**Institutional Review Board Statement:** Not applicable.

**Informed Consent Statement:** Not applicable.

**Data Availability Statement:** Available upon request from the corresponding author.

**Acknowledgments:** The authors are thankful to the Faculty of Agriculture, Kafrelsheikh University, Egypt for carrying out this work. New Valley University and Plant Pathology Research Institute, Agricultural Research Center are also thankfully acknowledged for the support provided for conducting this work. The Agricultural Research Center (ARC) in Egypt and the International Maize and Wheat Improvement Center (CIMMYT), are thankfully acknowledged for providing us the seeds of the inbred lines used in this study.

**Conflicts of Interest:** The authors declare no conflict of interest.

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
