# Peer review of "Molecular Genetic Diversity and Line × Tester Analysis for Resistance to Late Wilt Disease and Grain Yield in Maize"

_agronomy, doi:10.3390/agronomy11050898_

Round 1
Reviewer 1 Report
I made some comments to the authors: in a .pdf file with text highlighted in yellow and some comments; in a .doc file as well.

Author Response
Reviewer 1:
Comments
This study describes the performance of 30 Line × Tester crosses in two locations and their reaction to LWR in an artificially infested nursery. It is an interesting study, the experiments were well conducted and the manuscript is well written and has potential. However, I have some comments the authors should address before the manuscript can be considered for publication in the journal.
Re: We would like to thank the Reviewer for his time dedicated to our manuscript and presenting positive aspects in our manuscript. We highly appreciate the constructive criticisms which improved the manuscript.
- INTRODUCTION
Lines 58 to 61. To be exact, there are several other methods, although genetic resistance continues to be the most reliable one. Please replace by: “Chemical control of the disease is partially effective (Degani et al., 2018) or still under development (Tej et al., 2018). Thus, developing hybrids for resistance to LWD seems to be the best approach to reduce yield loss for smallholder farmers”. The references to be included are as follows:
Degani et al. 2018. PLOS ONE 13 (12), e0208353. DOI: 10.1371/journal.pone.0208353
Tej et al. 2018. EUR J PLANT PATHOL 152 (1): 249-265. DOI: 10.1007/s10658-018-1469-9
Re: The sentence has been modified and the references have been added (please see lines 60-65)
- MATERIALS AND METHODS
Line 95. Please delete “Zea mays L.”. This was already mentioned in line 49.
Re: “Zea mays L.” has been deleted (line 107)
Lines 105 and 121. The authors used TWC-368 as resistant and highly productive control in the yield and LWR experiments. However, for the 30 test-crosses they used SC-168, SC-176 and TWC-352 as parents (lines 101 and 102). Why none of them was included as control in the experiments given that all of them are highly productive? Conversely, why the TWC-368 hybrid (used as control) was not included as parent for the test-crosses? Please explain.
Re: The hybrid TWC-368 was used as a check hybrid because it is a newly developed hybrid and is recommended as a commercial high-yielding hybrid by the Egyptian Ministry of Agriculture and Land Reclamation. More information has been added (line 117).
Lines 118 and 119. “The other agronomic practices were conducted based on their recommendation.” Please reword.
Re: The sentence has been revised and rephrased (lines 130-133)
Lines 130-142. Please clarify which data were recorded in each of the experiments.
Re: Clarification has been added (lines 149-160)
Line 157. “Fresh leaves of 20-day-old seedlings were obtained from ...” Please be more specific: from how many plants? Were these parents sown in any of the two experiments? Were they sown in an independent experiment aimed at collecting leaf samples?
Re: Yes, it is independent pot experiment was performed to collect leaf samples, more clarification has been added (line 175-179)
- RESULTS
This section is structured according to statistics, which makes it difficult for the reader to follow and extract agronomic outcomes. I propose the authors to re-order the information (subheadings in blue) as follows so that the biological meaning is more easily understood:
Re: The results have been re-ordered and the tables have been splitted as suggested
3.1. Experiment of agronomic performance in two locations
Include here:
ANOVA and Line × Tester analysis for Agronomic Traits (current 3.1 section).
The corresponding information of mean performance (part of 3.3 section).
The corresponding information of General (GCA) and Specific (SCA) Combining Ability Effects (part of 3.4 section).
Current Table 1
Split Table 3 into two tables, each of them corresponding to one experiment: Values of DTS, EL, ED, NRPE, NKPR, HKW, GYPP (experiment of agronomic performance in two locations) and Values of Resistance to late wilt (experiment of LWR in the disease nursery). Insert here the corresponding information.
Split Table 4 similar to Table 3 and insert here the corresponding information.
Split Table 5 similar to Table 3 and insert here the corresponding information.
Re: Done
3.2. Experiment of resistance to late wilt
Include here:
ANOVA and Line × Tester analysis for Late Wilt Resistance (current 3.2 section).
The corresponding mean performance (part of 3.3 section).
The corresponding information of General (GCA) and Specific (SCA) Combining Ability Effects (part of 3.4 section).
Current Table 2 (correct footnote).
From the split Table 3, insert here the corresponding information.
From the split Table 4, insert here the corresponding information.
From the split Table 5, insert here the corresponding information.
Re: Done
3.3. (former 3.5) Interrelationship Among Measured Traits
In this section please state clearly why you decided to analyse the Interrelationship among traits from independent experiments (agronomic traits and disease resistance). Please support this decision with works by other authors.
Re: The Principal component analysis was applied on averages of the tested genotypes to determine the relationships among the evaluated traits particularly resistance to LWD under low and high N levels. References have been added (line 549)
3.4. Genetic analysis based on microsatellites
Merge here sections 3.6 (Microsatellites Based Polymorphism and 3.7. (Genetic distance (GD) and cluster Analysis)
Re: Done
Figure 4. If using the same legend in both panels, please place it in the middle of both and outside of panel A. Also, state what T1, T2 and T3 stand for.
Re: Done
3.5. Correlation Between Parental GD, Hybrid Performance and SCA
Similar to suggested for section 3.3, please state clearly why the authors decided to correlate GD, agronomic and reaction to disease performance and SCA (that is, handling together data from different experiments). Please support this decision with works by other authors.
Insert a Table with the data.
Re: The relationship between parental genetic distance and hybrid performance as well as SCA effects was tested to investigate the possibility of predicting hybrid performance and SCA effects in maize based on parental genetic distance using molecular markers. But in our results, the correlations between SSR marker-based GD and hybrid performance for GYPP and LWD were not significant. Similarly, GD had no significant correlation with SCA effects for GYPP (r=-0.08) and LWD (r=-0.15). Conversely, SCA effects of the evaluated crosses showed a strong correlation with the performance of F1 hybrid for both GYPP, (r= 0.65*) and LWD (r=0.43*). More clarification has been added (lines 497-499) and results of other works have been discussed (lines 646-653).
Also, throughout the Results section and when it applies, the authors must insert P values of significance and mean values of variables.
Re: p values and mean values have been added (lines 201-232, 276-285, 293-301, 319-325, 332-340, 348- 355)
- DISCUSSION
I have four main comments here.
Currently, the discussion includes many repetitions of the main results. I suggest eliminating those and instead discussing the results and formulating conclusions. I took the liberty to make comments to particular lines of the section (see uploaded). It is not that the writing is incorrect, but the readability of the section must be improved.
Re: The repeated sections have been deleted as suggested (lines 507-514, 559-560, 596-598)
Lines 446 to 454. The authors mention that variation in soil and climatic conditions between the two locations are the main cause for significant differences of agronomic traits. Thereafter, they relate to the LWR experiment, where differences between the two nitrogen levels were not found significant for resistance to LWD. The work by Ortiz-Bustos et al. (2019, https://doi.org/10.1111/ppa.13070) shows that environmental conditions can mask the effect of the pathogen on growth and productivity of maize. This could also happen when assessing the expression of resistance. The authors should discuss this point, considering that data from only one location and season were used for analysing LWR.
Re: More details have been discussed and the results of Ortiz-Bustos et al. (2019) has been added (lines 535-542)
Lines 535 to 536. “The SSR markers clearly grouped the parental genotypes into three main clusters which generally correspond with their origin.” In my opinion, this is incorrect or at least insufficiently clear. According to table S1, only two groups are differentiated within lines. The authors should discuss three findings here: i) the fact that L4 groups with L5, L6 and L7; ii) the differentiation of two clusters in lines L5 to L10; the finding that the controls L11, L12 and L13 group with L8 to L10. Also, when the authors refer to “source” (last column of tables S1) do they mean “provider”? Then no comparison of clusters to “source” can be made, since the term has no biological meaning. On the contrary, if the authors mean “geographic location” when they refer to source, then it could be possible to discuss on this (they mention the term “origin” in line 536. In any case, the authors must clarify which the origin of the lines is.
Re: The paragraph has been revised and rephrased (lines 635-642)
There is another point the authors should take into consideration. In a previous work by Sing and Siradhana (1990, Summa Phytopathologica 16:140-145), levels of 60 and 120 kgN/ha were considered as related to LW expression. Please discuss the results as compared to those by this work. Also, make reference to the implications of using high levels of N fertilization in the context of an agriculture that must be sustainable and friendly to the environment.
Re: The study of Sing and Siradhana (1990) and others have been discussed (lines 538-539)
If the authors address all these points, in my opinion, the manuscript could be considered for publication in the journal. Also, I would recommend the authors revise the abstract after corrections are included.
Best regards,
Reviewer 2 Report
Dear Authors
The manuscript is relatively well written, the experiments well performed and the results well scored. However, the molecular analyses had not any influence in the design of the remaining experiments.
The manuscript describes very common experiments, routinely carried out by other researchers with different objectives. No major sounding results, neither any kind of new contribution to the field (e.g. methodological) can be identified in the manuscript.
For that reason, I am of the opinion that it should not be accepted for publication in this journal.
Best Regards
Author Response
Reviewer 2:
Dear Authors
The manuscript is relatively well written, the experiments well performed and the results well scored. However, the molecular analyses had not any influence in the design of the remaining experiments. The manuscript describes very common experiments, routinely carried out by other researchers with different objectives. No major sounding results, neither any kind of new contribution to the field (e.g. methodological) can be identified in the manuscript.
For that reason, I am of the opinion that it should not be accepted for publication in this journal.
Re: We would like to thank the reviewer for his time dedicated to our manuscript and presenting positive aspects since it is well written, experiments well performed and the results well scored. We used line×tester mating design which is a common mating design carried out by other researchers with different objectives as described by the reviewer. In our case, we crossed between ten diverse maize inbred lines assembled from different origins and three high-yielding testers. The obtained thirty test-crosses were evaluated in two different locations for agronomic traits, besides, resistance to LWD was tested in disease nursery under artificial soil inoculation. We aimed to assess GCA and SCA of the evaluated local and introduced inbred lines for agronomic traits and resistance to LWD as well as to identify high-yielding hybrids with high resistance to LWD. LWD poses a major threat to maize production and accordingly, developing high-yielding and resistant hybrids is a vital approach to cope with this destructive disease. We have identified four superior hybrids with high-yielding and LWD resistance. We believe that our results introduce new promising hybrids with high-yielding ability and LWD resistance. On the other hand, the molecular analysis was performed to determine the parental genetic distance (GD) using SSR markers and investigating its relationship with hybrid performance and SCA effects.
Reviewer 3 Report
Please see attached file for comments. Eg., inadequate number of test environments

Author Response
Reviewer 3:
The study analyzes testcrosses and a standard check for resistance to late wilt disease (LWD), one of the devastating diseases of maize, under artificial soil infection. The study reported resistance sources having high yields and can form the basis of breeding for high yielding maize hybrids that are resistant to LWD. The paper is worth publishing after addressing some issues outlined below.
Re: We would like to thank the reviewer for his time dedicated to our manuscript.
Introduction
Ok.
Material and methods
-Well described. However, the yield trial was conducted in only two environments (i.e., two locations). At least an inclusion of another year/season (temporal factor) would help to estimate the environment effect which could make the estimates of the trait values closer to the expected values since these traits are highly influenced by the environment and GxE. I will leave this aspect to the judgment of the editor based on the journal’s standard on the number of test environments required. The authors have, however recommended multi-environment testing under the discussion section.
Re: We crossed between ten diverse inbred lines and three testers using line×tester mating design during the first growing season. In the second growing season, the obtained thirty test-crosses plus check hybrid were evaluated in two different locations for agronomic traits, in addition, resistance to LWD was tested in disease nursery under artificial soil inoculation. Our objectives were to assess GCA and SCA of the evaluated local and introduced lines for agronomic traits and resistance to LWD, identify high-yielding hybrids with high resistance to LWD and determine the parental genetic distance (GD) using SSR markers and its relationship with hybrid performance and SCA effects. In the discussion (line 554-556), we recommended further investigation across multi-environment trials for the identified 4 superior hybrids with high-yielding and LWD resistance to emphasize the consistency of their performance before commercialization.
-Late Wilt Trial: It is not clear to me whether “…the disease nursery under artificial soil inoculation” is a controlled environment (e.g., in the greenhouse) or not. If the genotypes were inoculated on the field as stated (“The field was infested…”), then only 1 season/year/location might not be enough for LWD resistance screening because disease incidence/severity/resistance is highly influenced by the weather conditions. This effect can be reduced by evaluating the genotypes in different locations/years/seasons unless the disease screening is carried out in a relatively controlled environment such as greenhouse or climate chambers. Stable genotypes across environments are crucial since a major objective of the authors was “… to identify high-yielding and LWD resistant hybrids”. The question that comes to mind is: are the authors aiming at identifying stable genotypes across different environments or not?.
Re: The used nursery is artificially infected field with the pathogen Magnaporthiopsis maydis and it is dedicated for evaluation. Annually, the used nursery is infected artificially by 4 clonal lineages of Magnaporthiopsis maydis to improve selection efficiency and distinguishing resistant and susceptible genotypes. The identified superior hybrids with high-yielding and LWD resistance are re-evaluated in the next step through standard protocol of the breeding program to emphasize their performance and stability.
-Also I suggest the use of ”infected” (fungus/pathogen) to replace “infested” Pg3, L125.
Re: “infested” has been replaced by “infected” (line 138)
-Page 3, L138: Please change trail to trial.
Re: “trail” has been replaced by “trial” (line 157)
Results
-Well presented. However, at Pg 7 L229, footnotes for Table 2; there are definitions for some abbreviations e.g. DTS, EL etc. which are not found in this table 2. Please, this MUST be corrected.
Re: The footnotes have been deleted (line 329)
-Pg8 L281: GYPP and grain yield/plant= GYPP: grain yield/plant
Re: has been corrected (line 314)
Discussion
Pg15, L479: “Developing maize hybrids have…”, please consider changing “have“ to having; “resistant” to resistance
Re: The sentence has been corrected as suggested